# Protein Ingestion in Reducing the Risk of Late-Onset Post-Exercise Hypoglycemia: A Pilot Study in Adolescents and Youth with Type 1 Diabetes

**DOI:** 10.3390/nu15030543

**Published:** 2023-01-20

**Authors:** Nirubasini Paramalingam, Barbara L. Keating, Tarini Chetty, Paul A. Fournier, Wayne H. K. Soon, Joanne M. O’Dea, Alison G. Roberts, Michael Horowitz, Timothy W. Jones, Elizabeth A. Davis

**Affiliations:** 1Department of Endocrinology and Diabetes, Perth Children’s Hospital, Perth, WA 6009, Australia; 2Children’s Diabetes Centre, Telethon Kids Institute, University of Western Australia, Perth, WA 6009, Australia; 3Department of Sport Science, Exercise and Health, School of Human Sciences, University of Western Australia, Perth, WA 6009, Australia; 4CRE in Translating Nutritional Science to Good Health, Faculty of Health and Medical Sciences, School of Medicine, University of Adelaide, Adelaide, SA 5000, Australia; 5Division of Paediatrics, within the Medical School, University of Western Australia, Perth, WA 6009, Australia

**Keywords:** type 1 diabetes, protein, LOPEH, glucagon, moderate intensity exercise, hypoglycemia

## Abstract

Dietary protein causes dose-dependent hyperglycemia in individuals with type 1 diabetes (T1D). This study investigated the effect of consuming 50 g of protein on overnight blood glucose levels (BGLs) following late-afternoon moderate-intensity exercise. Six participants (3M:3F) with T1D, HbA1c 7.5 ± 0.8% (58.0 ± 8.7 mmol/mol) and aged 20.2 ± 3.1 years exercised for 45 min at 1600 h and consumed a protein drink or water alone at 2000 h, on two separate days. A basal insulin euglycemic clamp was employed to measure the mean glucose infusion rates (m-GIR) required to maintain euglycemia on both nights. The m-GIR on the protein and water nights during the hypoglycemia risk period and overnight were 0.27 ± 043 vs. 1.60 ± 0.66 mg/kg/min (*p* = 0.028, r = 0.63) and 0.51 ± 0.16 vs. 1.34 ± 0.71 mg/kg/min (*p* = 0.028, r = 0.63), respectively. Despite ceasing intravenous glucose infusion on the protein night, the BGLs peaked at 9.6 ± 1.6 mmol/L, with a hypoglycemia risk period mean of 7.8 ± 1.5 mmol/L compared to 5.9 ± 0.4 mmol/L (*p* = 0.028) on the water night. The mean plasma glucagon levels were 51.5 ± 14.1 and 27.2 ± 10.1 ng/L (*p* = 0.028) on the protein and water night, respectively. This suggests that an intake of protein is effective at reducing the post-exercise hypoglycemia risk, potentially via a glucagon-mediated stimulation of glucose production. However, 50 g of protein may be excessive for maintaining euglycemia.

## 1. Introduction

The benefits of exercise in healthy individuals and individuals living with type 1 diabetes (T1D) are well known. However, the fear of hypoglycemia continues to be a barrier in the uptake of exercise by individuals living with T1D [1]. In individuals living with T1D, moderate-intensity exercise (MOD) performed in the late afternoon increases the risk of overnight late-onset post-exercise hypoglycemia (LOPEH) [2,3,4]. Indeed, McMahon et al. found that an additional 10 g of intravenous glucose, for a 70 kg person, was required to maintain euglycemia from 7 to 11 h after MOD performed in the late afternoon [4]. However, current strategies, such as insulin dose reduction and carbohydrate supplementation, are not always effective in reducing this risk [5], especially in individuals who are either on standard pump therapy or multiple daily injections. A study involving insulin pump users found no difference in the proportion of hypoglycemic events, with a 20% reduction in overnight basal insulin, or no basal reduction with the addition of terbutaline, as compared to no basal reduction [6]. 

Conversely, protein ingested alone or as part of a meal causes a late postprandial rise in blood glucose levels that require additional insulin to maintain euglycemia in individuals living with T1D [7,8,9,10]. Paterson et al. compared the postprandial glucose response after the ingestion of different amounts of pure glucose and pure protein in individuals with T1D who did not exercise in the day and who did not administer an insulin bolus for the glucose or protein ingested. They found that protein loads of up to 50 g did not result in significant postprandial glycemic excursions and that protein loads of 75 g and 100 g resulted in glucose excursions from 3 to 5 h after ingestion, which were comparable to that of 20 g of carbohydrate [11]. This suggests that consuming a protein snack in the evening after MOD performed in the late afternoon may provide another strategy for LOPEH prevention. 

There is also evidence that the ingestion of protein after exercise helps optimize muscle protein remodeling and repair [12] and that a protein ingestion pre-sleep may help optimize muscle protein synthesis [13]. Thus, ingesting protein post-exercise, for hypoglycemia prevention, may appeal to people with T1D who wish to optimize the training effect of their afternoon exercise since the ingestion of protein after exercise helps optimize muscle protein remodeling and repair. 

Thus, based on Patterson’s and McMahon’s work [4,11], we chose to investigate the effect of 50 g of whey protein as the minimum dose that would affect the post-exercise glucose requirements to maintain euglycemia in children and adolescents living with T1D who performed MOD in the late afternoon. We also chose to provide the protein supplement 3.25 h after exercise because this timing was deemed to be the most favorable at preventing nocturnal LOPEH based on the work of Patterson and McMahon [4,11]. Whey protein was chosen as the protein supplement as it is a rich source of amino acids and has the additional benefit of being an appetite suppressant [14]. 

The aim of this pilot study was to compare the difference in the mean intravenous glucose infusion rate to maintain overnight euglycemia following a bout of late afternoon MOD, with and without the consumption of 50 g of protein post-exercise, using a basal insulin glucose clamp protocol [4,15]. We hypothesized that the consumption of protein following late-afternoon MOD reduces the amount of intravenous glucose required to maintain euglycemia, particularly during the high-LOPEH risk period 7–11 h following late-afternoon MOD [4]. 

## 2. Materials and Methods

This pilot study was approved by the Child and Adolescent Health Service Human Research Ethics Committee, Western Australia. Participants and/or their parents provided informed consent to participate in the study.

### 2.1. Participants

Six participants with confirmed T1D duration > 1 year, who were hypoglycemia aware and otherwise healthy, participated in this study. Five participants were c-peptide negative (<0.05 mmol/L) pre-lunch, and one had a c-peptide level of 0.16 mmol/L. 

### 2.2. Study Design

This in-laboratory pilot study assessed the effect of consuming 50 g of protein in water vs. water alone on overnight glycemia and especially during the expected LOPEH risk period, following late-afternoon MOD, using a randomized counter-balanced study design. 

On two separate testing days, following a familiarization visit, participants performed MOD in the late afternoon followed by the intake of water only or a 50 g protein drink 3.25 h after exercise. The participants thus acted as their own controls, with female participants performing both of their exercise sessions in the same phase of their menstrual cycle. An established glucose clamp protocol, described in more details in earlier studies from our laboratory [4,15], was employed to maintain normal glucose levels via the intravenous infusion of glucose during both testing sessions. The intravenous rather than oral administration of glucose was adopted to allow the concentration of blood glucose to be rigorously matched between our experimental conditions before the intake of the protein/placebo supplement, and to measure precisely the amount of exogenous carbohydrate that is required to maintain stable glycemia after a bout of exercise. The difference in the post-test drink glucose infusion rates between the protein and water-only study nights was used to estimate the effectiveness of the protein drink in maintaining post-exercise euglycemia as compared to the water-only night. 

During the familiarization session, all participants were subjected to a $\dot{V}$O_2_ peak test, lactate threshold assessment, and the collection of baseline demographic and clinical measurements. On the two subsequent exercise testing days, one month apart, participants attended the laboratory at 1000 h. Hypoglycemia, exercise, alcohol, and caffeine were avoided in the 24 h preceding these testing days [15]. At 1200 h, participants had a standardized lunch and an intravenous insulin bolus (the meal and insulin bolus were matched on both days), with only water being permitted afterwards. At 1600 h, participants undertook 45 min of MOD on a stationary cycle ergometer (Corival, Lode BV, Groningen, The Netherlands) at 95% of their lactate threshold [4]. Following exercise, the participants rested for the rest of the study. At 2000 h, they consumed either 50 g of protein (56 g of Boomers whey protein isolate) in 125 mL of water (Pr-night) or 125 mL of water alone (Wa-night). The Boomers whey protein isolate contains nine essential amino acids (threonine, methionine, phenylalanine, histidine, lysine, valine, isoleucine, and tryptophan) and nine non-essential amino acids (serine, glycine, glutamic acid, proline, cysteine, alanine, tyrosine, arginine, and aspartic acid) [16]. Both test drinks were similarly flavored and artificially sweetened with Stevia and vanilla essence (equivalent to 1 g of glucose) as whey protein on its own was not palatable. 

The glucose clamp technique, utilized in previous LOPEH studies from our laboratory [4,15], was employed to maintain the blood glucose levels (BGLs) at euglycemia (5.5–6.5 mmol/L). Insulin Lispro (Humalog, Eli Lilly) was infused at a constant rate equivalent to 45% of the participant’s usual total daily dose of insulin. A 20% (weight/volume) dextrose solution was titrated, based on two consecutive BGL readings taken every 15–30 min, to achieve target BGLs from the study’s start at 1100 h to 0545 h the following morning. Blood samples were collected at fixed time points for the measurement of serum free insulin, plasma glucagon, glucose-dependent insulinotropic peptide (GIP), and glucagon-like peptide-1 (GLP1).

### 2.3. Biochemical Analyses

Venous BGLs were monitored (YSI 2300STAT, Yellow Springs Instrument, Yellow Springs, OH, USA) every 15 minutes to guide the glucose infusion rate. The serum free insulin levels were analyzed by an enzyme-linked immunosorbent assay (Architect, Abbot Laboratories), plasma GIP and GLP-1 by radioimmunoassay (GLPIT-36HK, Millipore, Billerica, MA, USA), and plasma glucagon by radioimmunoassay (GL-32K, Millipore, Billerica, MA, USA).

### 2.4. Statistical Analyses

The difference in the substrate and hormone levels between the Pr-night and Wa-night were analyzed using Wilcoxon signed-rank (WSR) tests, with a significance accepted at *p* < 0.05. The primary outcome was the difference in the mean glucose infusion rate (m-GIR) required to maintain euglycemia (5.5–6.5 mmol/L) during the high-LOPEH risk period from 2345 h to 0345 h (4) on the Wa-night and Pr-night and following the ingestion of the test drink at 2000 h to 0545 h the following morning. The m-GIR effect size (r) was calculated using the z-statistic of the WSR. Data were analyzed using IBM SPSS Statistics V24.0 and were reported as the mean ± SD and/or median (IQR) and graphically as the mean ± SEM. 

## 3. Results

### 3.1. Participants

The participants (3M:3F) were aged 20.2 ± 3.1 years, with a duration of diabetes 11.1 ± 4.9 years, mean HbA1c of 7.5 ± 0.8% (58 ± 8.7 mmol/mol), BMI of 26.7 ± 5.0 kg/m^2^, and $\dot{V}$O_2_ peak of 32.5 ± 9.7 mL/kg/min. 

### 3.2. Exercise and Clamp Conditions

All participants completed 45 min of exercise at the same intensity (55.7 ± 14.1 %$\dot{V}$O_2_ peak vs. 55.6 ± 13.7 %$\dot{V}$O_2_ peak; *p* = 0.9, and 81.4 ± 30.8 W vs. 78.0 ± 28.0 W; *p* = 0.4), on both the Pr-night and Wa-night. On both testing sessions, the intravenous insulin infusion rates were identical (1.35 ± 0.14 U/h). The mean serum free insulin levels were 10.3 ± 1.7 pmol/L and 10.7 ± 3.6 pmol/L (*p* = 0.6) from pre-exercise to the end of the study of the Pr-night and Wa-night, respectively. The m-GIR during the exercise sessions of the Wa-night and the Pr-night were similar (*p* = 0.2), as were the mean BGLs (*p* = 0.5). The 50 g of whey protein consumed, following MOD, was equivalent to 0.64 ± 0.10 g/kg.

### 3.3. Glucose Infusion Rates and Blood Glucose Levels

The pre-drink m-GIR (*p* = 0.9) and mean BGLs (*p* = 0.3) between the two nights were similar. Following the consumption of the test drink, at 195 min post-exercise, the m-GIR (median [IQR]) during the LOPEH risk period were 0.27 ± 0.43 mg/kg/min (0.03 [0.00, 0.64] mg/kg/min) and 1.60 ± 0.66 mg/kg/min (1.47 [1.02, 2.20] mg/kg/min) on the Pr-night and Wa-night, respectively (*p* = 0.028, r = 0.63). The overnight m-GIR were 0.51 ± 0.16 mg/kg/min (0.43 [0.39, 0.69] mg/kg/min) on the Pr-night and 1.34 ± 0.71 mg/kg/min (1.30 [0.68, 1.82] mg/kg/min) on the Wa-night, (*p* = 0.028, r = 0.63) (Figure 1A,B). 

The mean BGLs during the LOPEH risk period on the Pr-night and Wa-night were 7.8 ± 1.5 mmol/L and 5.9 ± 0.4 mmol/L (*p* = 0.028). The overnight mean BGLs were 7.2 ± 1.0 mmol/L and 6.0 ± 0.4 mmol/L (*p* = 0.028) on the Pr-night and Wa-night, respectively. The BGLs on the Pr-night peaked at 9.6 ± 1.6 mmol/L (range 7.7–11.9 mmol/L) despite ceasing the intravenous glucose on the Pr-night for 380 ± 112 min. 

### 3.4. Hormone Responses

The post-drink hormone responses were higher (*p* = 0.028) on the Pr-night than the Wa-night, with plasma glucagon levels of 51.5 ± 14.1 ng/L vs. 27.3 ± 10.1 ng/L, GIP levels of 26.8 ± 7.2 pmol/L vs. 13.9 ± 5.1 pmol/L, and GLP-1 levels of 18.8 ± 2.2 pmol/L vs. 10.2 ± 4.7 pmol/L. On the Pr-night, the glucagon levels peaked at 82.6 ± 26.9 ng/L at 65 ± 31 min and returned to the pre-drink levels within six hours. This peak in the glucagon response preceded the peak in BGLs on the protein night (Figure 1C–F). 

## 4. Discussion

Despite the many published recommendations available to people living with T1D for the prevention of post-exercise hypoglycemia [5], there is evidence that LOPEH continues to be a major challenge. This study shows that the ingestion of dietary protein in the evening, following late-afternoon MOD, provides a novel strategy to reduce the risk of overnight hypoglycemia in adolescents and youths living with T1D and who are otherwise healthy. To our knowledge, this is the first study to show that the ingestion of protein post-exercise compared to water alone results in a sustained increase in glycemia, without a continuous infusion of intravenous glucose, particularly during the period of an increased risk of nocturnal hypoglycemia 7–11 h post-exercise [4].

This pilot study, involving adolescents and youths with a mean duration of diabetes of 11.1 ± 4.9 years, shows that the ingestion of 50 g of whey protein markedly reduced the m-GIR needed to prevent LOPEH. We found that when protein was ingested approximately three hours after 45 min of MOD, the m-GIR on the protein night was six-fold lower during the 4 h LOPEH risk period [4] and 2.5-fold lower overnight than the water-only night. 

We also found that the ingestion of 50 g of protein resulted in transient hyperglycemia overnight. This hyperglycemia occurred despite the complete cessation of intravenous glucose whilst trying to maintain euglycemia on the protein-intake night. In contrast, a continuous infusion of intravenous glucose was invariably required to maintain euglycemia when water alone was ingested. This observed protein-mediated increase in the BGL is consistent with previous studies reporting that the ingestion of protein in non-exercise and basal insulin conditions resulted in a sustained and dose-dependent elevation of the blood glucose levels in individuals with T1D [7,11]. 

This increase in glycemia following a protein ingestion is likely to result from the prompt and major, albeit transient, rise in plasma glucagon [17], as increases in plasma amino acids resulting from the digestion of protein are known to stimulate a glucagon secretion [14,18]. This is in keeping with the findings of Rocha et al. [19] as whey protein isolate is a rich source of glucogenic amino acids. In addition, the stimulation of the secretion of GIP by the K-cells of the small intestine, which is also known to be a glucagon secretagogue in euglycemic conditions [20,21,22], may have also contributed to the rise in plasma glucagon. We propose that the glucagon-mediated stimulation of hepatic glycogenolysis and gluconeogenesis [20,23] is a likely mechanism responsible for both the increase in glycemia and the accompanying reduction in the m-GIR. Of note, unlike healthy subjects and people with type 2 diabetes [14,18,20,22], amino acids, GIP, and GLP-1 are not insulinotropic in people living with T1D because of the loss of endogenous insulin secretion. This mechanism may also have contributed to the rise in glycemia after the ingestion of whey protein for about six hours and inclusive of the period of the increased risk of LOPEH. This substantial difference in both the m-GIR and BGL between the Pr-night and Wa-night was no longer observed from approximately eight hours after consuming the protein drink or water alone. This may be attributable to both the reduction in plasma glucagon levels and the reduced availability of dietary glucogenic amino acids as the digestion of protein was reaching its completion. 

A strength of this pilot study is the use of the glucose clamp protocol to quantify the m-GIR required to maintain euglycemia under individualized near-basal insulin levels, a condition that replicates real life scenarios and has been used effectively in previous studies investigating LOPEH [4,15]. Additionally, the use of this clamp protocol removes the effect of potentially confounding variables such as the ingestion of concurrent carbohydrates and variability in the bioavailability of the subcutaneous delivery of insulin by delivering glucose intravenously. A potential limitation of this pilot study is its small sample size, but significance was reached in the primary outcome of difference in the glucose requirements between the Pr-night and Wa-night as well as the hormone responses with the six participants. One must also be cautious when translating the findings of this study directly into clinical practice. The 50 g (0.64 ± 0.10 g/kg) dose of protein that we utilized in this study is large relative to the recommended dietary protein intake for young adults (0.75 to 0.84 g/kg/day) [24] and even for athletes in training (0.4 g/kg/meal; 3–4 times per day) [25]. 

However, having demonstrated in this pilot basal insulin glucose clamp study that the ingestion of 50 g of whey protein in the evening is more than sufficient to maintain overnight euglycemia after a bout of late afternoon MOD, further studies can now investigate the effect of smaller doses of protein which could also be individualized to the weight of the individual in free-living conditions. There is also a substantial variation in the amino acid profile of different dietary protein sources [26]. As such, the magnitude of the glucagon and glycemic response observed with whey protein in this study may therefore be different for other protein types with a different amino acid profile. Further studies could explore the effect of different types of protein or whole foods that contain the recommended amounts of macronutrients combined with additional protein. The timing of the ingestion of these foods to alleviate both the early and late phases of post-exercise hypoglycemia should also be explored.

## 5. Conclusions

Consuming protein in the evening, after late-afternoon moderate intensity exercise, may provide a novel and effective strategy to prevent late-onset post-exercise hypoglycemia. Real life studies are now required to define the overnight blood glucose responses to different doses and types of protein ingested following late-afternoon exercise. 

## Figures and Tables

**Figure 1 nutrients-15-00543-f001:**
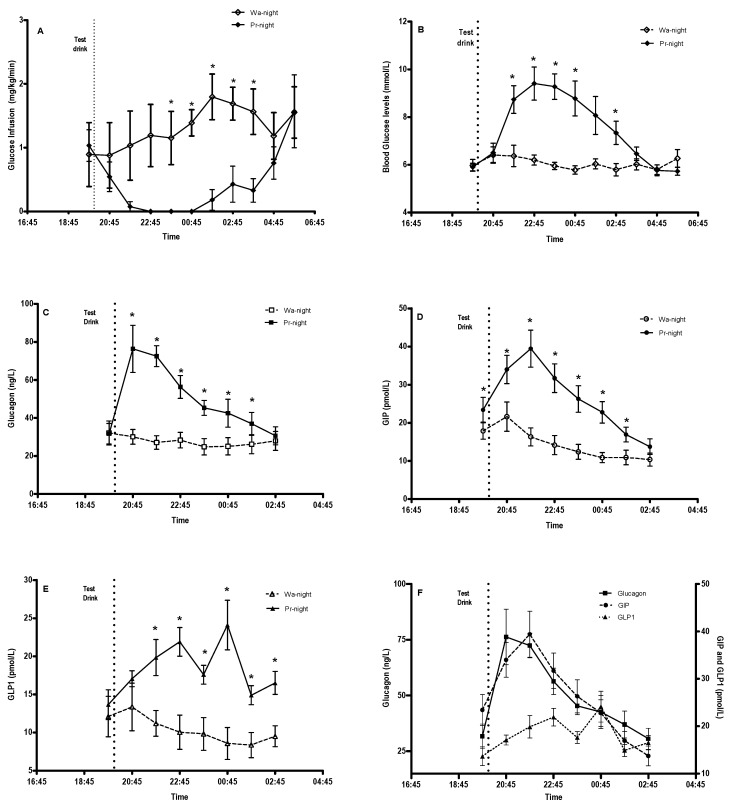
Glucose infusion rates, blood glucose levels, and hormone levels after completing 45 min of late-afternoon exercise at 1645 h and consuming water or 50 g whey protein at 2000 h. (**A**) Hourly mean glucose infusion rates, (**B**) blood glucose profile, (**C**) plasma glucagon response, (**D**) plasma GIP response, (**E**) plasma GLP-1 response, and (**F**) the association between glucagon, GIP, and GLP-1 on the Pr-night. * Significance between conditions accepted at *p *< 0.05.

## Data Availability

The data presented in this study are not publicly available due to ethical and privacy considerations.

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
