# Peer review of "Protein Ingestion in Reducing the Risk of Late-Onset Post-Exercise Hypoglycemia: A Pilot Study in Adolescents and Youth with Type 1 Diabetes"

_nutrients, 2023, doi:10.3390/nu15030543_

Round 1
Reviewer 1 Report
This study aims to compare the difference in the mean intravenous glucose infusion rate to maintain overnight euglycemia following a bout of late afternoon MOD, with and without the consumption of 50g of protein post-exercise. The authors concluded that protein intake is effective at reducing post-exercise hypoglycemia risk, potentially via a glucagon-mediated stimulation of glucose production.
This article is well written and very interesting.
However, several issues should be improved before the consideration for publication.
Major comments
1 The result of this study is interesting. However, it is unusual that type 1 diabetes patients do not take a meal particularly after intensive exercise. The patients are to be educated by medical stuff to prevent such meal skipping and consequent hypoglycemia.
In addition, the time point of taking protein (2000h), which is 3.25hr after the exercise, may be late. Meanwhile, if the effect of protein on blood glucose during sleep would be evaluated, the favorable time point of protein ingestion may be just before sleep.
Therefore, the conditions in this study hardly occur in the clinical practice in type 1 diabetes patients as well as type 2 diabetes patients.
I wonder that late onset post-exercise hypoglycemia can occur even after the consumption of dinner including usual amount of carbohydrate and protein. Please discuss such issues in the discussion.
2 The data of fasting C-peptide level in patients are informative because it reflects basal insulin level although very low levels. Considering the reproducibility in other study, such information is helpful.
3 It is better to provide the amino acid constituent of whey protein just in case.
4 I would like to know whether habitual high protein intake may aggravate blood glucose control in type 1 diabetes patients.
Reviewer 2 Report
The authors have investigated a practically very important field. Late-onset post-exercise hypoglycaemia (LOPEH) is a practical consequence of exercises in type 1 diabetes patients. The authors have performed experiments studying the possible way to avoid post-exercise hypoglycemia with a dietary element. The manuscript is well-written, the aim is clearly defined and the diagrams show the results proving the beneficial effects of post-exercise protein ingestion.
Round 2
Reviewer 1 Report
The manuscript has been improved according to the comments.